# Usefulness of Serum as a Non-Invasive Sample for the Detection of *Histoplasma capsulatum* Infections: Retrospective Comparative Analysis of Different Diagnostic Techniques and Quantification of Host Biomarkers

**DOI:** 10.3390/jof11060448

**Published:** 2025-06-12

**Authors:** L. Bernal-Martínez, P. De la Cruz-Ríos, R. Viedma, S. Gago, S. Ortega-Madueño, L. Alcazar-Fuoli, M. J. Buitrago

**Affiliations:** 1Mycology Reference Laboratory, National Center of Microbiology, Instituto de Salud Carlos III Majadahonda, 28222 Madrid, Spain; leticiabm@isciii.es (L.B.-M.); pauladelacruzr2@gmail.com (P.D.l.C.-R.); raulvb41@hotmail.com (R.V.); sara.gago-2@manchester.ac.uk (S.G.); sheilaortega@isciii.es (S.O.-M.); 2CIBERINFEC, ISCIII-CIBER de Enfermedades Infecciosas, Instituto de Salud Carlos III, 28029 Madrid, Spain

**Keywords:** *Histoplasma capsulatum*, histoplasmosis diagnosis, serum sample

## Abstract

Diagnosis of histoplasmosis is challenging. A rapid, sensitive, and specific method is essential. Serum is a non-invasive and easy sample to obtain in any hospital. The diagnostic accuracy of different techniques that use serum has been evaluated. Forty-one serum samples from patients with proven or probable histoplasmosis were analyzed. Different diagnostic techniques based on the detection of antibodies (ID Fungal Antibody System), antigens (Histoplasma GM EIA and Platelia^TM^ Aspergillus Ag), and DNA (“in-house” real-time PCR (RT-PCR) were tested and compared. Additionally, the quantification of cytokines and biomarkers related to histoplasmosis was performed. Global results from 27 samples in which all the tests were performed showed that the sensitivity of the Histoplasma GM EIA kit was 87.5% in patients with disseminated infection and HIV as an underlying disease; in immunocompetent (IC) patients, it was 54.5%. The detection of *Histoplasma* spp. with the ID Fungal Antibody System was positive in 90.9% of IC and in 62.5% of HIV patients. The Platelia-Asp kit had a low performance in both groups of patients (37.5% in HIV and 9% in non-HIV), and, finally, RT-PCR was better in immunosuppressed patients (44% in HIV vs. 27% in non-HIV). The combination of diagnostic techniques increased the detection of Histoplasma infection in inmunosupressed patients. Overall, patient groups infected with H. capsulatum (Hc) showed higher IL-8, IL-6, IL-1β, TNF-α, and IL-18 median values compared to non-Hc-infected controls. The effectiveness of diagnostic techniques on serum samples is highly influenced by the patient’s clinical presentation and underlying condition. Consequently, a thorough assessment of the patient’s clinical presentation and disease phenotype is crucial in selecting the most suitable diagnostic method.

## 1. Introduction

Histoplasmosis is a disease caused by the fungus *Histoplasma capsulatum* [1], which is a primary pathogen found in soil rich in organic material. Although this fungus has a cosmopolitan distribution, there are areas of high endemicity such as the Americas and Africa as well as other regions with medium-to-high endemicity including China [2] and Southeast Asia [3]. The disease is caused by the inhalation of microconidia found in soil enriched with organic material (i.e., guano). In the lungs, the spores convert to yeast form, being the main organ affected. In immunocompromised individuals, dissemination to other organs is frequent, especially in HIV/AIDS, where the infection can become progressively disseminated, affecting organs like the liver, spleen, bone marrow, and adrenal glands [4]. Despite the various presentations of the disease, disseminated histoplasmosis is the most frequent clinical presentation. Disseminated histoplasmosis is one of the most common opportunistic infections among people living with HIV in the Americas, and it is estimated to be responsible for 5% to 15% of AIDS-related deaths each year [5]. Histoplasmosis is also the imported mycoses most frequently reported outside these endemic regions [6,7]. Recently, based on its public health importance, global disease burden, and existing knowledge gaps, *H. capsulatum* is ranked as a high-priority pathogen in the WHO fungal priority pathogens list [8].

Diagnosis of histoplasmosis is challenging in both endemic and non-endemic regions. In endemic regions, the scarcity of tools for fast and accurate identification of the infection is a significant problem. In these regions, the disease is mainly associated with patients living with AIDS who develop the disseminated form of the disease, which can be fatal in the absence of rapid and appropriate treatment [9]. In non-endemic areas, in addition to the lack of diagnostic methods, a low index of suspicion must be added [10]. The gold standard for diagnosis of histoplasmosis is based on conventional laboratory assays using culture and histopathology. However, these assays have several limitations, including the need for high-level laboratory infrastructure for culture handling (biosecurity level 3), the need for highly trained laboratory personnel, variable assay analytical performance, and a long turn-around time for results. Alternative diagnostic methods are essential to achieve a rapid diagnosis as this has a significant impact on the patient’s outcome [11,12,13].

In recent years, a great effort has been made to develop methods based on the detection of different biomarkers. The selection of samples for the diagnosis of histoplasmosis is based on the clinical manifestation of the disease. In cases of acute pulmonary histoplasmosis, bronchoalveolar lavage (BAL), sputum, or lung biopsies are commonly utilized. For disseminated forms, samples such as bone marrow, peripheral blood (for cultures), lymph node aspirates, biopsies of affected organs, and urine (for antigen detection) are valuable. The choice of sample should be guided by the infection’s location and the patient’s immune status [14]. Serum is a non-invasive sample, easily obtainable, even in primary care centers, that can be used to detect antigens, antibodies, nucleic acids, and biomolecules to determine the evolution of an infection, making it a valuable sample.

The detection of antigens represented an important breakthrough in the early diagnosis of histoplasmosis; however, these tests have not been widely available in many regions until recently. Specifically, the *Histoplasma* GM ELISA (IMMY) test has demonstrated excellent performance and reproducibility in disseminated disease, but it has only been tested in urine samples until now [15]. Techniques based on the detection of antibodies, such as immunodiffusion or complement fixation, are commercially available and useful mainly for travelers from endemic regions; however, their sensitivity is moderate in immunosuppressed patients [16]. Techniques based on the detection of nucleic acids have shown to be very promising tools for rapid diagnosis, but there is a lack of commercial tests and consensus among laboratories [17].

In addition to pathogen-based diagnostic biomarkers, serum samples can reveal host-derived soluble mediators that provide insight into the mechanisms underlying the host’s failure to control the infection. Cytokines and chemokines play a critical role in shaping the immune response to *H. capsulatum*, with the outcome of infection being heavily influenced by the host’s early immune response [18]. Much of our current understanding of cytokine responses as diagnostic biomarkers originates from studies on various fungal infections; however, data specific to histoplasmosis remain limited. Previous works performed on bronchoalveolar lavage samples showed the differential role of certain cytokines in the local inflammatory processes of infection with *H. capsulatum*, *P. jirovecii*, or both [19] and the possibility to monitor certain cytokines levels for prognosis evaluation in patients [20].

The objective of this study was to evaluate the usefulness of serum as a non-invasive and easily obtainable sample in the diagnosis of histoplasmosis. For this purpose, the performance of techniques based on the detection of antibodies, antigens, and DNA was assessed. In addition, aiming to complement the diagnosis and predict prognosis, we analyzed the immunological response of patients using a panel of cytokines and other immunological biomarkers.

## 2. Materials and Methods

### 2.1. Patients and Clinical Samples

The Mycology Reference Laboratory of the National Center of Microbiology supports the National Health System in diagnosing endemic mycoses. This study was performed retrospectively using serum samples from 2005 to 2021 belonging to the collection of the laboratory and included in the ISCIII Biobank Collection. The samples were previously anonymized in compliance with Spanish law, and the ISCIII ethics committee approved the project (CEI PI 38_2019) date: 10 February 2020. The approval by the ethical committee for including the samples in the BioBank was referenced as CEI PI 71_2018, approval date: 5 November 2018.

In this study, a total of forty-one serum samples received for the routine diagnosis in the laboratory, from 40 patients with proven (30) and probable (10) histoplasmosis, classified based on criteria from EORTC/MSG were analyzed [21]. We aimed to include in the study patients with a range of clinical presentations from immunocompetent patients with mild symptoms, as well as patients with acute pulmonary histoplasmosis and other kinds of histoplasmosis and of course immunocompromised patients with disseminated infection.

Due to limitations in the available volume sample, not all tests were carried out in every sample. The specific number of samples used for each test is described for each diagnostic methodology. Among those, we were able to test all the diagnostic methodologies in 27 samples.

For the biomarker-cytokine study, additional samples from patients with probable infection were included in order to increase the number of patients and samples. For that purpose, we used 36 samples, of which 11 were additional samples and different from the samples initially used in the diagnostics studies. We also included eight serum samples from immunocompromised patients with no fungal infection according to the EORTC criteria [21].

### 2.2. Antigen Detection

Detection of *Histoplasma* GM (EIA test kit)

The clarus *Histoplasma* GM EIA test kit (IMMY Palex, Madrid, Spain) is validated and approved by the FDA for urine samples [15,22]. The serum needs to be pretreated to release diagnostically relevant antigens. These samples were pretreated according to the manufacturer’s recommendation as follows: a volume of 100 microliters of buffer supplied by the manufacturer was added to 300 microliters of each serum sample. The mixture was incubated at 120 °C for 6 min. Centrifugation was performed at 10,000 rpm for 10 min. The supernatant was used to carry out the indicated immunoassay technique. A cutoff of ≥0.20 ng/mL was used to determine positivity, according to the manufacturer’s recommendations.

Detection of *Histoplasma capsulatum* by Platelia^TM^
*Aspergillus* Ag (Bio-Rad)

A volume of 300 microliters of serum was used. Serum samples were pre-treated with heat in the presence of treatment solution, and then 50 µL of treated samples was added to the well, and the mixture was incubated with the conjugated antibody following the manufacturer’s recommendations. The optical density was measured at 450 nm with a reference at 620 nm. A cutoff of ≥0.5 was used to determine positivity.

### 2.3. Antibody Detection

Detection of *Histoplasma* antibodies by immunodiffusion by using the “ID Fungal Antibody System” kit (IMMY, Palex, Madrid, Spain) was used following the manufacturer’s instructions. A volume of 20 µL was placed in a well against a control antigen placed in another well in the diffusion medium (CleargelTM, Palex, Madrid, Spain) and allowed to diffuse outward into the medium. Precipitation bands were analyzed after 48 h.

### 2.4. DNA Detection

DNA extraction was performed using the QiAmp DNA Mini Kit (Qiagen, Werfen, Madrid, Spain) following the manufacturer’s instructions. Fifty microliters of elution buffer were used for elution.

The RT-PCR technique described by Gago et al. [23] was employed for the detection *H. capsulatum*. The assay is a multiplex real-time PCR that detects three fungal pathogen species (*Pneumocystis jirovecii*, *H. capsulatum,* and *Cryptococcus neoformans/Cryptococcus gatii*) and includes an internal control. The technique was standardized and validated in clinical samples. The target for *H. capsulatum* is the ITS 2 region of the ribosomal DNA (sequences of primers and probes are available in PCT/ES2009070340).

Four microliters of DNA from the clinical samples were used as template. All tests were performed in duplicate, including negative and positive controls for the PCR.

### 2.5. Biomarkers Quantification

A cytokine panel including IL-1β, IL-8, IL-17, IL-23, TNF-α, IL-6, IL-10, IL-18, and the soluble proteins PTX3 and sTREM1 were tested in serum samples using Luminex with the Bio-Plex equipment (Bio-Rad, CA, USA). For sample preparation, the commercial Human premixed Multi-Analyte kit (R&D Systems, MN, USA) was used, following the protocol recommended by the manufacturer. All cytokine determinations were performed in duplicate, and concentrations were reported in pg/mL.

### 2.6. Statistical Analysis

All data were analyzed with GraphPad Prism 9.2.0 software. To determine significant differences between the study groups, unpaired *t*-test were performed using the Mann–Whitney test. *p* values < 0.05 were considered statically significant.

The correlation between biomarkers among *H. capsulatum*-infected patients was explored using the Spearman correlation test.

## 3. Results

### 3.1. Patients and Clinical Samples

Most patients 80% (32/40) were immigrants from Latin-American and African countries, and the rest were travelers to endemic areas. Regarding underlying diseases, 20 patients had AIDS, 15 of whom presented with a disseminated histoplasmosis (DH), 3 with acute pulmonary histoplasmosis (APH), and 2 with a gastrointestinal form (GIH). A total of 19 were immunocompetent patients with different forms of the disease, 9 with APH or sub-acute pulmonary histoplasmosis (SAPH), 1 patient with mediastinitis, 1 with pyomyositis, 1 with chronic respiratory condition, 3 with rheumatoid arthritis, 1 patient with primary skin infection, and 3 non-HIV patients with no clinical information collected. Only one patient had another type of immunosuppression different from AIDS and presented a disseminated disease (Table 1).

### 3.2. Antigen Detection

The GM EIA test was performed on 33 samples. To test cross-reactivity with other closely related fungal species, a total of 25 serum from patients infected with species of *Aspergillus* spp. (8), *Candida* spp. (5), and other endemic species such as *Paracoccidiodes* spp. (7) and *Coccidiodes* spp. (5) were included as the control.

The sensitivity of the GM EIA test kit in immunosuppressed patients (IS), mostly with AIDS, was 94% (17/18) and in immunocompetent patients (IC) was 66% (10/15). Patients with disseminated disease were positive at 100% (13/13). Acute pulmonary histoplasmosis (APH) was detected at 75% (9/12), and gastrointestinal histoplasmosis (GIH) at 50% (1/2).

The Platelia^TM^
*Aspergillus* Ag was performed on 37 samples but demonstrated low performance in both groups: 42% (8/19) in IS and 16% (3/18) in IC. The best results were obtained in patients with DH 53% (8/15) vs. patients with APH (1/12) and was negative for chronic and mild disease.

### 3.3. Antibody Detection

The detection of antibodies using the ID fungal antibody system was performed in 40 samples and was positive in 31 out of 40 (77.5%), 95% (18/19) of ICs and 62% of IS (13/21). Patients with disseminated disease were positive at 56.25% (9/16), APH 85% (11/13), and GIH 66% (2/3); mild symptoms 100% (3/3).

### 3.4. DNA Detection

The PCR technique was performed in 36 samples, and results were moderated for the detection of DNA. The overall sensitivity was 36% (13/36), being more useful for IS patients at 50% in HIV (10/20) vs. 19% in non-HIV patients (3/16). Patients with disseminated infections had the best results 62.5% (10/16).

### 3.5. Comparative Analysis of Tools Used for Diagnostic

All methods were performed in 27 samples from 26 patients. Individual results are presented in Table 2, and aggregated results are summarized in Table 3.

We explored the potential of using combinations of techniques to improve diagnostic performance (Table 4). The combination of two diagnostic tests gave interesting results for immunosuppressed patients, highlighting their increased power of detecting *Histoplasma* spp. infections.

### 3.6. Biomarker Quantification

Since immune response is expected to be different between immunossupressed (IS) or immuno-competent (IC) individuals, patients were grouped into three categories: IC_Hc (Immuno-competent with *H. capsulatum* infection), IS HIV_Hc (Immunossupressed with HIV and *H. capsulatum* infection), and IS_non-Hc (Immunossupressed with non *H. capsulatum* infection).

Overall, patient groups infected with *H. capsulatum* (IC-Hc and IS HIV-Hc groups) showed higher IL-8, IL-6, IL-1B, TNF-α, and IL-18 median values compared to non-Hc-infected controls, although no significance was reached for some of the analyses. The IS HIV-Hc group showed the highest median values with statistical differences for IL-6, IL-1Β, TNF-α, and IL-18 (Figure 1).

IL-17A and IL-23 were analyzed, with no significant difference values among groups.

The anti-inflammatory cytokine IL-10 results also showed higher values for the Hc-infected groups, and statistical differences were found for both groups when compared to the IS_non-Hc patients.

Regarding the soluble biomarkers PTX3 and sTREM1 (Figure 2), a significantly higher median value for PTX3 was found when IS HIV_Hc and IC_Hc groups were compared. Compared to the IS_non-Hc group, PTX3 values were also higher, although no significance was reached. TREM1 showed significantly higher median values of IS HIV_Hc patients against IS_non-Hc individuals.

To further investigate biomarkers and disease conditions, we performed a correlation analysis of serum biomarkers among patients that developed a disseminated histoplasmosis and patients with a non-disseminated form of the disease (Figure 3). Disseminated patients demonstrated positive correlations between certain biomarkers. The strongest correlations were observed where IL-8 correlated well with IL-1β, IL-17A, and sTREM1, and the strongest correlation was found between IL-1β and sTREM1. In the group of the non-disseminated form of the disease, we found positive correlation for IL-8 and IL-18, and IL-23 with IL-1β and IL-10, which showed correlation with IL-6.

## 4. Discussion

The diagnosis of histoplasmosis classically relies on culture and microscopy. The isolation of the fungus on culture or the visualization of yeast in tissues from clinical samples are considered the gold standard, but these methods have well-known limitations [24]. Recently, new methods based on the detection of antigens, antibodies, and DNA have been developed, leading to an improvement in diagnosis [25]. However, the diagnosis of histoplasmosis remains challenging due mainly to the lack of the availability of some of these methods in certain areas, as well as due to the different clinical presentations that make suspicion and detection difficult [26]. Several studies published in the literature address different approaches to the diagnosis of histoplasmosis, including serological methods, which are useful but have limited sensitivity, particularly in immunocompromised patients [26]. Antigen detection in urine and serum is highly sensitive in disseminated histoplasmosis cases [27]. Combining antigen detection with serology results, which is effective in cases involving the pulmonary system and CNS [28,29] or with molecular methods, such as PCR or nested PCR, enhances diagnostic accuracy [30]. However, molecular techniques may show higher specificity or faster turnaround times, aligning with the push rapid diagnostics [31], but these methods show variability results due to a lack of standardization and reliance on “in-house” methods [32,33,34]. Other emerging diagnostic tools like next-generation sequencing (NGS) and metagenomic approaches are promising but not routine. A recently published study provides information on fungal biomarkers such as galactomannan (Platelia Aspergillus) and β-D-glucan (Fungitell), which could serve as alternatives in settings where commercial antigen tests are not available [35]. The possible divergent findings of results with different techniques must be interpreted carefully, considering the study population, their immunosuppression, the environment, whether endemic or not, the types of samples, and methodological differences.

Our initial hypothesis questioned the suitability of serum as an efficient sample for the rapid diagnosis of histoplasmosis across diverse patient populations and using diagnostic techniques. Serum appears to be the most accessible and easily collected biological sample in clinical settings. We evaluated the effectiveness of various serum-based diagnostic methods for histoplasmosis. Additionally, we performed cytokine profiling in different patient groups to identify potential disease biomarkers and assess their diagnostic value.

This retrospective study utilized remaining serum samples from routine diagnostic procedures. As histoplasmosis is not endemic in our region, the overall sample size was limited. An even smaller subset of samples had sufficient volume to perform all intended tests. To enhance the sample size for cytokine analysis, we included additional specimens, which represent a limitation of this study. Further results will be incorporated as more data become available.

Regarding the methods employed, three commercial methods were used, an immunodifussion method for the detection of antibodies, an EIA specific for the detection *Histoplasma* galactomannan antigen, and the Platelia galactomannan for *Aspergillus*. The reason for using the latter relies on the fact that *H. capsulatum* and other endemic fungi cross-react with the Platelia test for *Aspergillus* spp., and this test has indeed been helpful in those laboratories without access to *Histoplasma*-specific tests [36]. As far as we know, this is the first study that used serum samples for the GM-EIA Kit as previous works used urine samples [37]. For the detection of nucleic acids, an “in-house” technique previously described [23] and validated was performed.

Concerning the results for the 27 samples in which all tests were performed, the best results were obtained in immunocompetent patients with the immunodiffusion technique, which detected 90.9% of all sera, showing its usefulness. However, this technique does not allow for discrimination between past or active infections. Moreover, the seroconversion requires 4 to 8 weeks to be able to detect antibodies in serum. In patients with immunosuppression (HIV+) and disseminated disease, the *Histoplasma* galactomannan antigen detection technique (S 87.5%) obtained a suitable performance. This technique has the limitation of not being used very widely in non-endemic regions since it is not cost-effective due to the low number of cases in these regions. The positive Platelia *Aspergillus* results were associated mainly with disseminated disease; however, the moderate sensitivity indicates that it is not a good option to rule out histoplasmosis. In immunocompetent patients, the sensitivity was very low, indicating that it should not be used. Regarding the RT-PCR results, the moderate sensitivity obtained suggested that the amount of circulating DNA in blood is very low. It would be necessary to develop alternative extraction methods that use large volumes of serum to improve the performance of the technique since PCR has the advantage of being fast, easy, specific, and increasingly cheaper.

The combination of techniques seems to be an adequate option to overcome the limitations of the different techniques in serum samples. Improved results were obtained by combining RT-PCR and ID. The combination of these two techniques is easy to implement in laboratories and can be especially useful in non-endemic regions since they can be easily performed for a low number of samples in the clinical laboratory. ID and *Aspergillus* Platelia also could be optimal in non-endemic regions. A workflow proposal has been presented to guide in the diagnosis of histoplasmosis in microbiology laboratories (Appendix A). However, it is a preliminary proposal, and more data should be added to confirm these results.

Focusing on the study of cytokines, to date, the immune response to histoplasmosis has been mostly studied using animal or in vitro infection models [38]. Following *H. capsulatum* deposition in the lungs, effective infection control relies on a strong proinflammatory immune response [18]. Key cytokines such as IL-6 and IL-23 promote T-cell differentiation, with Th1 cells, characterized by the production of IFN-γ, TNF-α, and IL-1β, playing a central role in activating macrophages to suppress fungal growth. The Th17 response also contributes significantly through the secretion of cytokines including IL-17, IL-21, IL-22, TNF-α, and IL-6. Among these, IL-17 is particularly important, as it induces the expression of proinflammatory cytokines and chemokines that facilitate the recruitment and activation of macrophages and neutrophils. IL-18 is another critical cytokine in the early immune response to *H. capsulatum.* It is involved in activating Th1 cells, which are important for controlling fungal infections and may contribute to the formation of granulomas, which are protective structures in the lungs. IL-8 (CXCL8), which is secreted primarily by epithelial cells in the lung, plays a key role in neutrophil recruitment and activation during infection. It has been seen that yeasts of *H. capsulatum* can directly stimulate IL-8 secretion by A549 epithelial cells, suggesting the fungus actively modulates the local inflammatory environment in the lung [39]. The immune response is also regulated by anti-inflammatory cytokines, particularly IL-10, which limits excessive inflammation. For instance, IL-10-deficient models increase levels of protective cytokines and are associated with faster fungal clearance. Although anti-inflammatory cytokines can impair fungal clearance, they are essential for preventing the immune-mediated pathology by maintaining immune balance during histoplasmosis.

In this context, it has been described that in the murine lung infected with *H. capsulatum*, elevated concentrations of the cytokines IL-1β, TNF-α, IL-6, IL-17, IL-23, IL-12, IFN-γ, IL-4, and IL-10 were detected [40]. Additionally, in humans, increased concentrations of cytokines IL-1β and TNF-α, along with others such as IFN-γ, IL-18, IL-17A, IL-33, IL-13, and CXCL8, have been observed [19]. In our work, compared to other groups, IS HIV_Hc patients showed the highest median values for IL-8, IL-6, IL-1B, TNF-α, IL-18, PTX3, and sTREM1 (Figure 1 and Figure 2). TNF- α and IL-1β contribute to the generation of protective immunity against infection with *H. capsulatum* [18]. In our study, TNF-α and IL-1β together with IL-6 and IL-18 showed the highest values, with statistical differences for the group of HIV patients and Hc suggesting that these cytokines play a role in the local inflammatory processes of histoplasmosis in HIV patients. Our results also showed elevated levels of the soluble pattern recognition molecule pentraxin 3 (PTX3), as well as the soluble form of the triggering Receptor Expressed on Myeloid Cells 1 (sTREM1) in the IS HIV_Hc group. These two components of the immune response have shown a role against fungal disease with, for example, elevated levels in serum samples of hematological patients with invasive aspergillosis [41]. The analysis for exploring associations between different biomarkers demonstrated a strong positive correlation between several cytokines in patients with disseminated histoplasmosis. This finding suggests that these cytokines play a role in the human immune response to *Histoplasma* spp., with immune responses differing between immunocompetent and immunocompromised individuals.

It might appear difficult to use the information provided by host biomarkers such as cytokines to improve the diagnosis of fungal infections and even more in the context of histoplasmosis and immunosuppression associated with HIV. However, very little has been investigated on that, so increasing our knowledge about the role of soluble mediators that occur during these infections will provide information about host–pathogen interactions and how the host fights against the pathogen. This knowledge can be therefore used to improve diagnosis in a particular group of patients and to develop therapeutic strategies limiting the progressive invasive disease.

## 5. Conclusions

Serum samples are suitable for rapid diagnosis of histoplasmosis, but the performance of available techniques depends on the patient’s clinical presentation. The GM EIA kit appears to be a reasonable selection for diagnosing disseminated histoplasmosis. The ID technique is valuable for detecting *H. capsulatum* antibodies in immunocompetent patients. The RT-PCR technique, however, exhibited moderate performance in serum and should be used in combination with other techniques. The combination of techniques seems to be the best approach to obtain excellent performance. Elevated IL-1β, TNF- α, IL-18, and PTX3 levels in HIV patients with histoplasmosis could serve as a potential predictive biomarker for poor prognosis and disseminated disease development in these individuals. To assess the specificity and confirm the utility of these techniques in histoplasmosis diagnosis, further studies with larger patient cohorts, control subjects, and individuals with other infections are necessary.

## Figures and Tables

**Figure 1 jof-11-00448-f001:**
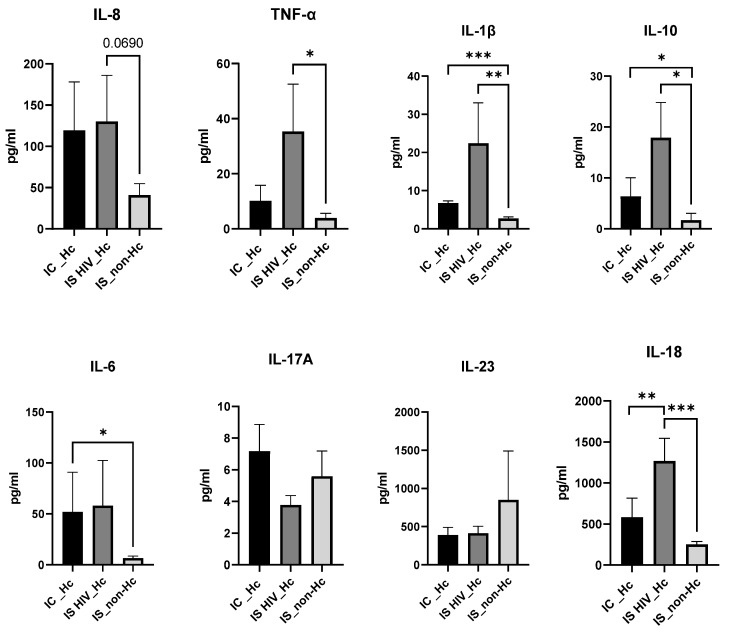
Cytokine levels (pg/mL) from serum samples measured by Bioplex Multi-Analyte assays. Statistical significance was tested using the Mann–Whitney test (pairwise comparisons). *p* < 0.05 was considered statistically significant; all statistically significant comparisons are shown according to * *p* < 0.05, ** *p* < 0.01, and *** *p* < 0.001.

**Figure 2 jof-11-00448-f002:**
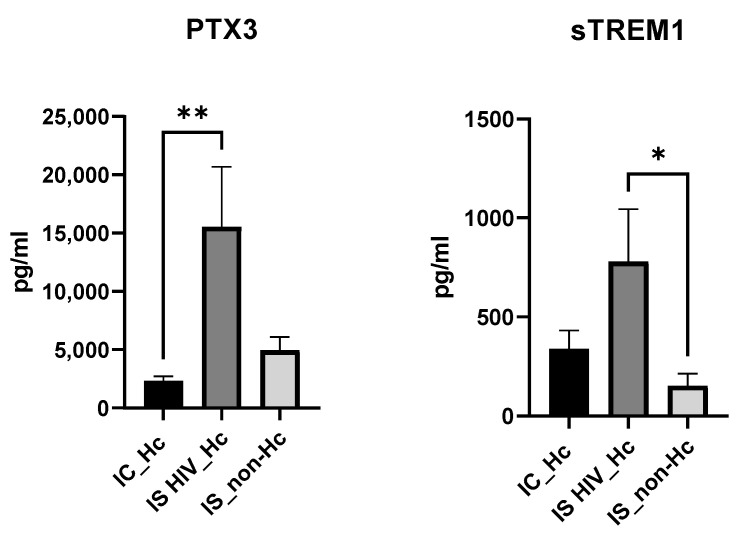
PTX3 and sTREM1 levels (pg/mL) from serum samples measured by Bioplex Multi-Analyte assays. Statistical significance was tested using the Mann–Whitney test (pairwise comparisons). *p* < 0.05 was considered statistically significant; all statistically significant comparisons are shown according to * *p* < 0.05 and ** *p* < 0.01.

**Figure 3 jof-11-00448-f003:**
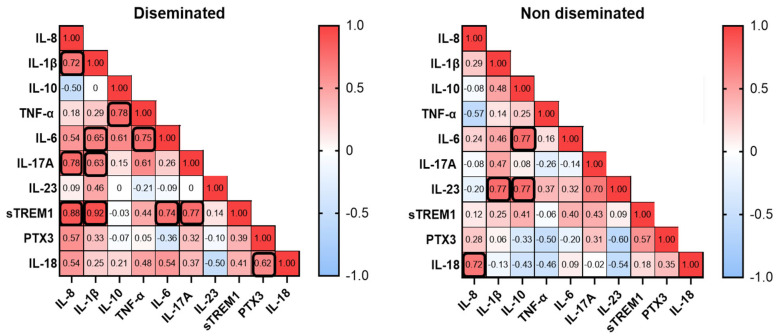
Correlation analysis of serum biomarkers. The scale represents the Spearman rho value; red squares correspond to the maximum positive correlation and blue squares to the maximum negative correlation. Numbers represent the correlation values, and black bold squares indicate statistical significance. *p* < 0.05 was considered statistically significant.

**Table 1 jof-11-00448-t001:** Clinical characteristics of 40 proven or probable histoplasmosis patients.

Characteristics	Number
**Age, mean, years**	40
**Gender:**	
Male (%)	62
Female (%)	38
**Originating from an endemic region** (Nicaragua, Colombia, Guatemala, Costa Rica, Bolivia-via, Uruguay, Ecuador, Peru, Nigeria, Senegal, Venezuela, Brazil, Paraguay, Dominican Republic, Pana-ma, Argentina, El Salvador, Equatorial Guinea, Mexico, and Ghana)	32
**Travelers**	8
**Underliying condition:**	
HIV/AIDS	20
Inmunosupressed non-HIV	1
None	19

**Table 2 jof-11-00448-t002:** Results for 27 sera samples in which all diagnostic tests were performed.

Patient nº	Immunosuppression	RT-PCR	ID	Platelia Asp	GM EIA	Clinical Data	Classification
1	No	+	+	−	+	APH	Proven
2	No	+	+	−	+	APH	Proven
3	No	−	+	+	+	APH	Probable
7	No	−	+	−	−	APH	Proven
7 *	No	−	+	−	+	APH	Proven
28	No	−	−	−	−	APH	Proven
29	No	−	+	−	−	APH	Proven
33	No	−	+	−	−	CPH	Probable
35	No	−	+	−	−	APH	Probable
37	No	−	+	−	+	ACPH	Probable
38	No	+	+	−	+	ND	Probable
4	Yes	+	−	−	+	DH	Proven
8	Yes	+	+	+	+	DH	Proven
9	Yes	−	−	−	+	DH	Proven
10	Yes	+	−	+	+	DH	Proven
11	Yes	−	−	−	−	GIH	Proven
13	Yes	−	+	−	+	DH	Proven
16	Yes	+	−	+	+	DH	Proven
17	Yes	−	+	−	+	DH	Proven
18	Yes	+	−	+	+	DH	Proven
19	Yes	−	+	−	+	GIH	Proven
20	Yes	+	+	+	+	DH	Proven
30	Yes	+	+	−	+	DH	Proven
32	Yes	−	+	+	+	DH	Proven
34	Yes	−	+	−	+	DH	Proven
36	Yes	−	+	−	−	APH	Proven
40	Yes	−	+	−	+	SAPH	Probable

Positive (+), negative (−). APH: acute pulmonary histoplasmosis; DH: disseminated histoplasmosis; GIH: gastrointestinal histoplasmosis; CPH: chronic pulmonary histoplasmosis; SAPH: sub acute pulmonary histoplasmosis: ND: no data. * same patient.

**Table 3 jof-11-00448-t003:** Summary of results for each test in 27 samples.

Immunosuppression	RT-PCR	ID	Platelia Asp	GM EIA
YES 16/27 (59%)	7/16 (44%)	10/16 (62.5%)	6/16 (37.5%)	14/16 (87.5%)
NO 11/27 (41%)	3/11 (27%)	10/11 (90.9%)	1/11 (9%)	6/11 (54.5%)
TOTAL 27	10/27 (37%)	20/27 (74%)	7/27 (26%)	20/27 (74%)

**Table 4 jof-11-00448-t004:** Results of combining diagnostic tests in serum samples.

Immunosuppression	RT-PCR +ID	RT-PCR + Platelia Asp	RT-PCR + GM EIA	ID + Platelia Asp	ID + EIA GM	Platelia Asp + GM EIA
YES 16/27 (59%)	13/16 (81.25%)	8/16 (50%)	13/16 (81.25%)	12/16 (75%)	15/16 (93.7%)	14/16 (87.5%)
NO 11/27 (41%)	10/11 (90.9%)	3/11 (27.2%)	6/11 (54.5%)	10/11 (90.9%)	10/11 (90.9%)	6/11 (54.5%)
TOTAL 27	23/27 (85.1%)	11/27 (40.7%)	19/27 (70.3%)	12/27 (81.4%)	25/27 (92.6%)	20/27 (74%)

## Data Availability

The original contributions presented in the study are included in the article; further inquiries can be directed at the corresponding author.

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
