# Peer review of "Usefulness of Serum as a Non-Invasive Sample for the Detection of Histoplasma capsulatum Infections: Retrospective Comparative Analysis of Different Diagnostic Techniques and Quantification of Host Biomarkers"

_jof, 2025, doi:10.3390/jof11060448_

Round 1
Reviewer 1 Report
Introduction
The authors should briefly describe the pathogenesis of histoplasmosis to justify the rationale for the selected diagnostic samples.
Materials and Methods
2.1 Patients and clinical Samples
What is the protocol approval number from the ISCIII Ethics Committee? This number must be stated in the manuscript.
What inclusion and exclusion criteria were applied to the study population? Please provide these details.
The description of sample usage is unclear. The text mentions 40 samples, but not all were tested in every assay, and additional samples were included later for cytokine analysis. This lack of standardization complicates the interpretation of results. I suggest restricting the analysis to the 27 samples tested across all assays.
What are the sociodemographic characteristics of the patients? This information should be described or referenced, as it can provide important context for understanding the results.
2.4 DNA Detection
Please expand the RT-PCR description. What are the oligonucleotide sequences for ITS2? Were the results obtained via absolute quantification? What parameter was measured?
Discussion
Please add reference in the first paragraph.
Please discuss the current findings in the context of existing literature on histoplasmosis diagnostics. Are the results consistent or divergent? If divergent, how can these differences be explained?
Discuss the roles of the evaluated cytokines in Histoplasma infection. Why is the detection of certain cytokines relevant in the context of infection? This should be described in more detail.
What is the clinical applicability of the tests evaluated? Could they be implemented in routine diagnostic workflows, or are they intended for research use?
The limited number of samples is a limitation. In addition, the lack of sociodemographic data limits interpretation. Including this information could strengthen the study’s conclusions.
L48 - The article cited is over a decade old. Is disseminated histoplasmosis still the most common AIDS-defining disease and leading cause of death in HIV-positive patients today? Please update this information with more recent literature.
L179 - Please italicize Aspergillus
Table 2 - I suggest replacing “positive” and “negative” with the symbols “+” and “–”
L271–275 - These sentences describe methodological procedures and should be moved to the Methods section.
Author Response
- The authors should briefly describe the pathogenesis of histoplasmosis to justify the rationale for the selected diagnostic samples.
We agree with the reviewer, two new paragraphs have been included in the introduction to clarify this point:
“The disease is caused by the inhalation of microconidia found in soil enriched with organic material (ie: guano). In the lungs, the spores convert to the yeast form, being the main organ affected. In immunocompromised individuals, dissemination to other organs is frequent, especially in (HIV/AIDS) where the infection can become progressive disseminated affecting organs like liver, spleen, bone marrow, adrenal glands, etc.”
“The selection of samples for the diagnosis of histoplasmosis is based on the clinical manifestation of the disease. In cases of acute pulmonary histoplasmosis, bronchoalveolar lavage (BAL), sputum, or lung biopsies are commonly utilized. For disseminated forms, samples such as bone marrow, peripheral blood (for cultures), lymph node aspirates, biopsies of affected organs, and urine (for antigen detection) are valuable. The choice of sample should be guided by the infection's location and the patient's immune status.”
Materials and Methods
2.1 Patients and clinical Samples
- What is the protocol approval number from the ISCIII Ethics Committee? This number must be stated in the manuscript.
Part of this information was already included in the Institutional Review Board Statement section. It has now been more detailed by adding the approval numbers and the approval dates. The same information is also included in the Materials and Methods section:
“This study was performed retrospectively using serum samples from 2005 to 2021 and belongs to the collection of clinical specimens of the Mycology Reference Laboratory included in the ISCIII Biobank Collection. The samples were previously anonymized in compliance with Spanish law and the ISCIII ethics committee approved the project (CEI PI 38_2019) date: 10/02/2020. The approval by the Ethical committee for including the samples in the BioBank is the number CEI PI 71_2018 approved date: 5/11/2018.”
- What inclusion and exclusion criteria were applied to the study population? Please provide these details.
Our laboratory supports the National Health System for the diagnosis of endemic mycoses. We included samples received for the diagnosis and classified as histoplasmosis probable or proven by the EORTC criteria. We aimed to include in the study patients with a range of clinical presentations from immunocompetent patients with mild symptoms, as well as patients with acute pulmonary histoplasmosis and other kinds of histoplasmosis and, of course, immunocompromised patients with disseminated infection. A paragraph has been included in the text in section Patients and clinical samples.
- The description of sample usage is unclear. The text mentions 40 samples, but not all were tested in every assay, and additional samples were included later for cytokine analysis. This lack of standardization complicates the interpretation of results. I suggest restricting the analysis to the 27 samples tested across all assays.
We appreciate the reviewer’s comment and we would like to provide a detailed explanation of our sample selection process in response. We are not an endemic region for histoplasmosis; therefore, the number of available samples is limited. As the national reference laboratory, we support the National Health System in diagnosing endemic mycoses. For this study, we used the remaining material from routine diagnostic procedures performed in our laboratory. This limited the volume of sample available and, consequently, our ability to perform all tests on every specimen.
The initial diagnostic technique used—immunodiffusion—is the method routinely applied to serum samples. Subsequent tests were conducted depending on the amount of remaining sample, which was constrained by the initial volume received, prior usage, and the potential need for retesting.
Given that our region is not endemic for histoplasmosis, the sample size was inherently small. We agree with the reviewer that focusing on the subset of 27 samples could provide a clearer picture. However, we believe that limiting the analysis to only these may lead to the loss of valuable data. In response to the reviewer’s suggestions, we have revised abstract, results , and discussion to highlight the findings from the 27 samples. Additionally, we have included a section on the study’s limitations in the discussion.
- What are the sociodemographic characteristics of the patients? This information should be described or referenced, as it can provide important context for understanding the results.
Our center is not a hospital, but a referral center, and we do not have access to patients' medical records. Most immunosuppressed patients with AIDS are immigrants from Latin American countries, while immunocompetent patients are mostly travelers. Among the data that we had available, the average age was 40 years old and 62 % are men and 38% women. We have added this information together with the countries of origin grouping them in table 1.
- 4 DNA Detection
Please expand the RT-PCR description. What are the oligonucleotide sequences for ITS2? Were the results obtained via absolute quantification? What parameter was measured?
PCR Sequences for the primers and probe for H. capsulatum are included in patent PCT/ES2009070340. They are accessible on Google Patents and can be used as long as it is not for commercial purposes.
This paragraph has been included in DNA detection section:
“The assay is a multiplex PCR that detects three fungal pathogens species (Pneumocystis ji-rovecii, H. capsulatum and Cryptococcus neoformans/Cryptococcus gatii) and includes internal control. The technique was standardized and validated in clinical samples. The target for H. capsulatum is the ITS 2 region of the ribosomal DNA (Sequences of primers and probes available in PCT/ES2009070340)”.
Discussion
- Please add reference in the first paragraph.
Done as requested, these references have been added.
- Please discuss the current findings in the context of existing literature on histoplasmosis diagnostics. Are the results consistent or divergent? If divergent, how can these differences be explained?
We agree, and a paragraph discussing this issue has been incorporated into the manuscript.
“Several studies published in the literature address different approaches to the diagnosis of histoplasmosis, including serological methods which are useful but have limited sensitivity, particularly in immunocompromised patients 26. Antigen detection in urine and serum is highly sensitive in disseminated histoplasmosis cases 27. Combining antigen detection with serology results which is effective in cases involving the pulmonary system and CNS 28, 29 or with molecular methods, such as PCR or nested PCR, enhances diagnostic accuracy 30. However, molecular techniques may show higher specificity or faster turnaround times, aligning with the push rapid diagnostics 31, but these methods show variability results due to a lack of standardization and reliance on “in house” methods 32-34. Other emerging diagnostic tools like Next Generation Sequencing (NGS) and metagenomic approaches are promising but not routine. A recently published study provides information on fungal biomarkers such as galactomannan (Platelia Aspergillus) and β-D-glucan (Fungitell), which could serve as alternatives in settings where commercial antigen tests are not available 35. The possible divergent findings of results with different techniques must be interpreted carefully, considering the study population, their immunosuppression, the environment, whether endemic or not, the types of samples, and methodological differences”.
- Discuss the roles of the evaluated cytokines in Histoplasma infection. Why is the detection of certain cytokines relevant in the context of infection? This should be described in more detail.
We thank the reviewer for this comment and we have now tried to explain it better. We have made some changes in the introduction (lines 88 - 92) as well as in the discussion (lines 352 - 373).
- What is the clinical applicability of the tests evaluated? Could they be implemented in routine diagnostic workflows, or are they intended for research use?
We fully agree with the reviewer, the results can be implemented in the microbiology laboratory. In fact, the results obtained when combining techniques are very interesting. We have included a suggested workflow as supplementary figure 1. Although it is preliminary, we believe that it can be very useful for laboratories, especially in non-endemic regions.
- The limited number of samples is a limitation. In addition, the lack of sociodemographic data limits interpretation. Including this information could strengthen the study’s conclusions.
We think we have answered this question in previous comments.
Detail comments
- L48 - The article cited is over a decade old. Is disseminated histoplasmosis still the most common AIDS-defining disease and leading cause of death in HIV-positive patients today? Please update this information with more recent literature.
Done as requested
- L179 - Please italicize Aspergillus
Done as requested
- Table 2 - I suggest replacing “positive” and “negative” with the symbols “+” and “–”
Done as requested
- L271–275 - These sentences describe methodological procedures and should be moved to the Methods section.
We agree with the reviewer, this paragraph has been eliminated from the discussion.
Reviewer 2 Report
I read your work with enthusiasm, since, as you rightly point out, histoplasmosis diagnosis requires more cost-effective techniques for any laboratory, as well as less invasive samples. Your study is very interesting and, overall, clear.
My main criticism is regarding:
- Title: I suggest modifying it as it reflects nothing on the biomarkers-cytokines study.
- From 40 serum samples, only 27 are actually useful for comparing diagnostic methods. What is the relevance of mentioning the 40 samples? Why not limit it to the 27 samples that could be used for the method comparison? Why didn't you consider the availability of the sample volume required for all tests as an inclusion criterion?
- Why wasn't the biomarkers-cytokines study conducted with the same 27 samples? Can't this be considered a limitation of the study?
- Please, indicate the date of approval by the ethics committee
- Line 52 remove the period and separate the words: “H.capsulatum.”
- Line 104 Change Sub to sub
- Line 138 remove one period
- Line 334 Change Histoplasmosis to histoplasmosis
- Table 1 I suggest you delete Table 1, since it does not contain any additional information to the text. If you decide to keep it, change the title to: Clinical Characteristic of 40 proven or probable histoplasmosis patients
- Table 2 This goes down at the bottom of the table. APH: 196 Acute Pulmonary Histoplasmosis; DH: Disseminated Histoplasmosis; GIH: 197 gastrointestnal histoplasmosis; CPH: Cronic Pulmonary Histoplasmosis; SAPH: Sub 198 Acute Pulmonary Histoplasmosis: ND: No Data. *same patient.
- Table 2 I suggest you presenting the data pooled, first the results from immunosuppressed patients and then those from immunocompetent patients, to facilitate comparison.
- Lines 113-114, 117-122 This represents a very important limitation of the study. Please discuss it in the corresponding section.
- Figure 2 where is the figure 2?
- Lines 286-287 Delete “The samples were pretreated as recommended by the manufacturer´s.” This is already mentioned in Materials and Methods
Author Response
I read your work with enthusiasm, since, as you rightly point out, histoplasmosis diagnosis requires more cost-effective techniques for any laboratory, as well as less invasive samples. Your study is very interesting and, overall, clear.
My main criticism is regarding:
Title: I suggest modifying it as it reflects nothing on the biomarkers-cytokines study.
Done as requested. The title has been changed.
- From 40 serum samples, only 27 are actually useful for comparing diagnostic methods. What is the relevance of mentioning the 40 samples? Why not limit it to the 27 samples that could be used for the method comparison? Why didn't you consider the availability of the sample volume required for all tests as an inclusion criterion?
We appreciate the reviewer’s comment and we would like to provide a detailed explanation of our sample selection process in response. We are not an endemic region for histoplasmosis; therefore, the number of available samples is limited. As the national reference laboratory, we support the National Health System in diagnosing endemic mycoses. For this study, we used the remaining material from routine diagnostic procedures performed in our laboratory. This limited the volume of sample available and, consequently, our ability to perform all tests on every specimen.
The initial diagnostic technique used—immunodiffusion—is the method routinely applied to serum samples. Subsequent tests were conducted depending on the amount of remaining sample, which was constrained by the initial volume received, prior usage, and the potential need for retesting.
Given that our region is not endemic for histoplasmosis, the sample size was inherently small. We agree with the reviewer that focusing on the subset of 27 samples could provide a clearer picture. However, we believe that limiting the analysis to only these may lead to the loss of valuable data. In response to the reviewer’s suggestions, we have revised abstract, results , and discussion to highlight the findings from the 27 samples. Additionally, we have included a section on the study’s limitations in the discussion.
- Why wasn't the biomarkers-cytokines study conducted with the same 27 samples? Can't this be considered a limitation of the study?
The biomarker-cytokine study was the last one performed using the remaining available samples. As mentioned earlier, the number of serum samples was limited. Although we received other types of samples—such as BALs, biopsies, and sputum—we aimed to maintain the original focus of the study, which was based on serum samples. However, in order to increase the sample size for the cytokine analysis, we included other specimens with sufficient volume to carry out the analysis.
Detail comments
- Please, indicate the date of approval by the ethics committee
Part of this information was already included in the Institutional Review Board Statement section. In any case, now it has been more detailed by adding the approval numbers and the approval dates. The same information is also include in the Materials and Methods section:
“This study was performed retrospectively using serum samples from 2005 to 2021 and belonging to the collection of clinical specimens of the Mycology Reference Laboratory included in the ISCIII Biobank Collection. The samples were previously anonymized in compliance with Spanish law and the ISCIII ethics committee approved the project (CEI PI 38_2019) date: 10/02/2020. The approval by the Ethical committee for including the samples in the BioBank is the number CEI PI 71_2018 approved date: 5/11/2018”
- Line 52 remove the period and separate the words: “H.capsulatum.”
Done as requested
- Line 104 Change Sub to sub
Done as requested
- Line 138 remove one period
Done as requested
- Line 334 Change Histoplasmosis to histoplasmosis
Done as requested
- Table 1 I suggest you delete Table 1, since it does not contain any additional information to the text. If you decide to keep it, change the title to: Clinical Characteristic of 40 proven or probable histoplasmosis patients.
We would prefer to keep it, so according to the reviewer we have changed the title. We have added demographic characteristics of the patients.
- Table 2 This goes down at the bottom of the table. APH: 196 Acute Pulmonary Histoplasmosis; DH: Disseminated Histoplasmosis; GIH: 197 gastrointestnal histoplasmosis; CPH: Cronic Pulmonary Histoplasmosis; SAPH: Sub 198 Acute Pulmonary Histoplasmosis: ND: No Data. *same patient.
Done as requested.
- Table 2 I suggest you presenting the data pooled, first the results from immunosuppressed patients and then those from immunocompetent patients, to facilitate comparison.
We agree with the reviewer. Done as requested.
- Lines 113-114, 117-122. This represents a very important limitation of the study. Please discuss it in the corresponding section.
We are not an endemic region for histoplasmosis; therefore, the number of available samples is limited. As the national reference laboratory, we support the National Health System in diagnosing endemic mycoses. For this study, we used the remaining material from routine diagnostic procedures performed in our laboratory. This limited the volume of sample available and, consequently, our ability to perform all tests on every specimen.
The initial diagnostic technique used—immunodiffusion—is the method routinely applied to serum samples. Subsequent tests were conducted depending on the amount of remaining sample, which was constrained by the initial volume received, prior usage, and the potential need for retesting.
The biomarker-cytokine study was the last one performed using the remaining available samples. As mentioned earlier, the number of serum samples was limited. Although we received other types of samples—such as BALs, biopsies, and sputum—we aimed to maintain the original focus of the study, which was based on serum samples. However, in order to increase the sample size for the cytokine analysis, we included other specimens with sufficient volume to carry out the analysis.
Given that our region is not endemic for histoplasmosis, the sample size was inherently small. We agree with the reviewer that focusing on the subset of 27 samples could provide a clearer picture. However, we believe that limiting the analysis to only these may lead to the loss of valuable data. In response to the reviewer’s suggestions, we have revised abstract, results, and discussion to highlight the findings from the 27 samples. Additionally, we have included a section on the study’s limitations in the discussion.
- Figure 2 where is the figure 2?
Figure 2 is now in the document.
- Lines 286-287 Delete “The samples were pretreated as recommended by the manufacturer´s.” This is already mentioned in Materials and Methods.
Done as requested
Reviewer 3 Report
The article addresses a relevant topic with important applicability and potential impact in the scientific field. However, the contextualization needs improvement due to the limited number of scientific references cited. This weakness directly affects the quality of the discussion, particularly in terms of comparing findings, formulating interpretations of the results, and relating them to previously published studies.
The manuscript addresses a relevant and timely topic with potential clinical applicability. The use of serum as a non-invasive sample for histoplasmosis diagnosis is of significant interest. However, the study presents methodological limitations, which should be addressed to improve the manuscript’s clarity, rigor, and overall impact.
The study's rationale, along with the relevance of the topic, its originality, and clinical applicability, should be given greater emphasis and a more thorough discussion.
The objective of the study is unclear. I suggest rewriting it clearly and concisely.
What is the study hypothesis?
Begin the methodology section by stating the study design, including general information about the research conducted. Please provide essential missing information, such as the ethics committee approval number, primary and secondary variables, and potential confounding factors.
How was informed consent obtained from the individuals whose samples were used in the research?
Include the sample size calculation.
Transfer the sample characteristics information to the results section, such as: “Most patients 80% (32/40) were immigrants from Latin-American and African countries, and the rest were travelers to endemic areas. Regarding underlying diseases, 20 patients had AIDS, fifteen…”.
Remove from the methodology section any information that should be presented in the results section, such as: “To test cross-reactivity with other closely related fungal species, a total of 25 serum from patients infected with species of Aspergillus spp. (8), Candida spp. (5), and other endemic species such as Paracoccidioides spp. (7) and Coccidioides spp. (5) were included as controls.”
Ensure consistent formatting of Aspergillus in italics throughout the text.
In the description of the tests performed, it is frequently mentioned that the manufacturer's recommendations were followed. However, this approach makes the methodology less attractive to readers and may hinder comprehension.
Provide a brief summary of the manufacturers' recommendations to enhance clarity and flow.
Clarify the method referenced in the sentence “The RT-PCR technique described by Gago et al.”
Include a brief justification, with references, explaining why the tests were performed in duplicate rather than triplicate.
In the results section, it would be relevant to add information obtained from the medical histories (anamneses) of the individuals whose samples were used. Would it be possible to access this information? If so, include sociodemographic characteristics.
Add a descriptive statistical analysis for Figures 1, 2, and 3.
Figure 2 is missing.
Replace the uppercase "P" with lowercase "p" when reporting p-values.
In the discussion section, address the study hypothesis and explain whether it was accepted or rejected, including the rationale.
The opening sentence of the conclusion repeats a generic idea (“depends on clinical presentation and technique”) that is already detailed in the following statements. This could be condensed for clarity.
The section discussing biomarkers appears abruptly between the diagnostic techniques and the suggestion for future studies. Reorganizing this part would improve the overall flow.
Some expressions could be revised for greater formality and precision, such as “suitable choice” and “valuable for detecting.”
Author Response
Major comments
- The article addresses a relevant topic with important applicability and potential impact in the scientific field. However, the contextualization needs improvement due to the limited number of scientific references cited. This weakness directly affects the quality of the discussion, particularly in terms of comparing findings, formulating interpretations of the results, and relating them to previously published studies.
We agree with the reviewer and new references have been added. According to that, the discussion section has been modified discussing recent works.
Detail comments
- The manuscript addresses a relevant and timely topic with potential clinical applicability. The use of serum as a non-invasive sample for histoplasmosis diagnosis is of significant interest. However, the study presents methodological limitations, which should be addressed to improve the manuscript’s clarity, rigor, and overall impact.
We agree with the reviewer but, given that our region is not endemic for histoplasmosis, the sample size was inherently small. We have focused on the subset of 27 samples to provide a clearer picture. We have revised the abstract, results, and discussion to highlight the findings from the 27 samples. Additionally, we have included a section on the study’s limitations in the discussion.
- The study's rationale, along with the relevance of the topic, its originality, and clinical applicability, should be given greater emphasis and a more thorough discussion.
We fully agree with the reviewer. In fact, the positive results obtained for immunosuppressed patients when combining techniques are very encouraging. This is something we are currently evaluating with the goal of implementing it in our reference laboratory for the diagnosis of Histoplasmosis in HIV patients. We have included a suggested workflow as supplementary figure 1. Although it is preliminary, we believe that it can be very useful for laboratories, especially in non-endemic regions.
- The objective of the study is unclear. I suggest rewriting it clearly and concisely.
Done as requested. The objective has been stated more clearly in the last paragraph of the introduction (lines 98 - 103).
- What is the study hypothesis?
The hypothesis is the following question: is serum sample useful for the diagnosis in all kinds of patients and all techniques available? The hypothesis has been added in the discussion.
- Begin the methodology section by stating the study design, including general information about the research conducted. Please provide essential missing information, such as the ethics committee approval number, primary and secondary variables, and potential confounding factors.
Information regarding the ethics committee approval number has been added:
“The Mycology Reference Laboratory of the National Center of Microbiology supports the National Health System in diagnosing endemic mycoses. This study was performed retrospectively using serum samples from 2005 to 2021 belonging to the collection of the Laboratory and included in the ISCIII Biobank Collection. The samples were previously anonymized in compliance with Spanish law and the ISCIII ethics committee approved the project (CEI PI 38_2019) date: 10/02/2020. The approval by the Ethical committee for including the samples in the BioBank was referenced as CEI PI 71_2018 approval date: 5/11/2018 “.
- How was informed consent obtained from the individuals whose samples were used in the research?
Our laboratory supports the National Health System for the diagnosis of endemic mycoses. For this study, the remaining sample from routine diagnostic work in the laboratory was used. Samples were anonymized and included in the biobank of the institution. The ethical committee approved the use of the samples in the project. The numbers of the Ethical Committee are CEI PI 38_2019 and CEI PI 71_2018.
- Include the sample size calculation.
Although the ideal is to calculate the sample size to perform a more complete study, in practice, the number of samples available is limited, and in this case, a significant sample volume was required, which further limited the study. Finally, it must be considered that we are not in an endemic area and do not have many samples for each kind of patient. Taking into account the results of this work, future studies will be carried out with a larger number of samples.
- Transfer the sample characteristics information to the results section, such as: “Most patients 80% (32/40) were immigrants from Latin-American and African countries, and the rest were travelers to endemic areas. Regarding underlying diseases, 20 patients had AIDS, fifteen…”.
Done as requested. This part has been included in Results “Patients and Clinical samples” section.
- Remove from the methodology section any information that should be presented in the results section, such as: “To test cross-reactivity with other closely related fungal species, a total of 25 serum from patients infected with species of Aspergillus spp. (8), Candida spp. (5), and other endemic species such as Paracoccidioides spp. (7) and Coccidioides spp. (5) were included as controls.”
Done as requested. This paragraph has been included in Results section
- Ensure consistent formatting of Aspergillus in italics throughout the text.
Done as requested
- In the description of the tests performed, it is frequently mentioned that the manufacturer's recommendations were followed. However, this approach makes the methodology less attractive to readers and may hinder comprehension.
Provide a brief summary of the manufacturers' recommendations to enhance clarity and flow.
Done as requested. A summary has been added for each technique in Materials and Methods.
- Clarify the method referenced in the sentence “The RT-PCR technique described by Gago et al.”
This paragraph has been included in DNA detection section:
The assay is a multiplex Real Time PCR that detects three fungal pathogens species (Pneumocystis jirovecii, H. capsulatum and Cryptococcus neoformans/Cryptococcus gatii) and includes internal control. The technique was standardized and validated in clinical samples. The target for H. capsulatum is the ITS 2 region of the ribosomal DNA (Sequences of primers and probes available in PCT/ES2009070340).
- Include a brief justification, with references, explaining why the tests were performed in duplicate rather than triplicate.
We are not an endemic region for histoplasmosis; therefore, the number of available samples is limited. As the national reference laboratory, we support the National Health System in diagnosing endemic mycoses. For this study, we used the remaining material from routine diagnostic procedures performed in our laboratory. It is very difficult to perform assays in triplicate, but controls were included in all test to verify results.
- In the results section, it would be relevant to add information obtained from the medical histories (anamneses) of the individuals whose samples were used. Would it be possible to access this information? If so, include sociodemographic characteristics.
Our center is not a hospital, but a referral center, and we do not have access to patients' medical records. Most immunosuppressed patients with AIDS are immigrants from Latin American countries, while immunocompetent patients are mostly travelers. Among the data that we have, the average age was 40 years old and 62 % are men and 38% women. We have added this information together with the countries of origin grouping them in table 1.
- Add a descriptive statistical analysis for Figures 1, 2, and 3.
Statistical analysis is included at the bottom of the each figure.
- Figure 2 is missing.
Figure 2 is now in the document.
- Replace the uppercase "P" with lowercase "p" when reporting p-values.
Done as requested.
- In the discussion section, address the study hypothesis and explain whether it was accepted or rejected, including the rationale.
The hypothesis initially formulated was: Can serum samples be effectively used for the diagnosis of histoplasmosis across all patient groups and diagnostic techniques? This hypothesis has been incorporated into the Discussion section.
- The opening sentence of the conclusion repeats a generic idea (“depends on clinical presentation and technique”) that is already detailed in the following statements. This could be condensed for clarity.
We have changed the opening of the conclusions.
- The section discussing biomarkers appears abruptly between the diagnostic techniques and the suggestion for future studies. Reorganizing this part would improve the overall flow.
We thank the reviewer for this comment and we have now tried to explain it better. We have made some changes in the introduction (lines 88 - 92) as well as in the discussion (lines 352 - 373).
- Some expressions could be revised for greater formality and precision, such as “suitable choice” and “valuable for detecting.”
Revised as requested.
Round 2
Reviewer 1 Report
NA
L331 - Please replace histoplasma with "Histoplasma"
Reviewer 2 Report
Thank you for considering the suggestions
Thank you for considering the suggestions